# Ultra-wideband-responsive photon conversion through co-sensitization in lanthanide nanocrystals

Zhao Jiang [1], Liangrui He[1], Zhiwen Yang[1], Huibin Qiu [2], Xiaoyuan Chen [3], Xujiang Yu [1]✉ & Wanwan Li [1]✉

Distinctive upconversion or downshifting of lanthanide nanocrystals holds promise for biomedical and photonic applications. However, either process requires high-energy lasers at discrete wavelengths for excitation. Here we demonstrate that co-sensitization can break this limitation with ultrawide excitation bands. We achieve co-sensitization by employing $Nd^{3+}$ and $Ho^{3+}$ as the co-sensitizers with complementary absorptions from the ultraviolet to infrared region. Symmetric penta-layer core-shell nanostructure enables tunable fluorescence in the visible and the second near-infrared window when incorporating different activators ($Er^{3+}$, $Ho^{3+}$, $Pr^{3+}$, and $Tm^{3+}$). Transient spectra confirm the directional energy transfer from sensitizers to activators through the bridge of $Yb^{3+}$. We validate the features of the nanocrystals for low-powered white light-emitting diode-mediated whole-body angiography of mice with a signal-to-noise ratio of 12.3 and excitation-regulated encryption. This co-sensitization strategy paves a new way in lanthanide nanocrystals for multidirectional photon conversion manipulation and excitation-bandwidth-regulated fluorescence applications.

Smart control of photon conversion in lanthanide nanocrystals (LnNCs) is an important topic owing to its valuable scientific and technological implications on bioimaging[1–3], cancer treatment[4,5], optical sensing[6], information storage and security[7], and so on. The abundant electronic energy levels of trivalent lanthanide ions ($Ln^{3+}$) offer the possibility for controllable regulation of photon conversion in an ultrawide range spanning from the ultraviolet (UV) to infrared[8–10]. However, the intrinsic parity-forbidden 4f-4f electronic transition of $Ln^{3+}$ leads to ultralow molar extinction coefficients $<10\ cm^{-1}\ M^{-1}$ with linear absorption behaviors[11,12]. Hence costly high-energy lasers at specific wavelengths, instead of cost-effective and readily available light sources, are required for the excitation of LnNCs.

Sensitization has been demonstrated to be highly efficient in tuning the absorption and photon conversion behaviors of LnNCs, and currently reported strategies including $Ln^{3+}$-$Ln^{3+}$ sensitization[13,14], transition metal sensitization[15], semiconductor-matrix sensitization[16,17], dye sensitization[18,19], and multicomponent nanostructure sensitization[20]. Among these, the $Ln^{3+}$-$Ln^{3+}$ sensitization plays a pivotal role due to its capability of imparting particular absorption and emission on a considerable wavelength scale (Supplementary Fig. 1a). The scope of this $Ln^{3+}$-$Ln^{3+}$ sensitization has recently been extended by Wang and Zhou et al. by introducing a $Gd^{3+}$ or $Yb^{3+}$ energy-migratory mediator in the sensitizer-activator pair (Supplementary Fig. 1b), which allows multiplexed photon upconversion under 980 or 1530 nm excitation,

[1]State Key Lab of Metal Matrix Composites, School of Materials Science and Engineering, Zhangjiang Institute for Advanced Study, Shanghai Jiao Tong University, 800 Dongchuan Road, Shanghai 200240, P. R. China. [2]State Key Laboratory of Metal Matrix Composites, Frontiers Science Centre for Transformative Molecules, School of Chemistry and Chemical Engineering, Shanghai Jiao Tong University, Shanghai 200240, P. R. China. [3]Yong Loo Lin School of Medicine and Faculty of Engineering, National University of Singapore, Singapore 117597, Singapore. ✉e-mail: yuxuj1017@sjtu.edu.cn; wwli@sjtu.edu.cn

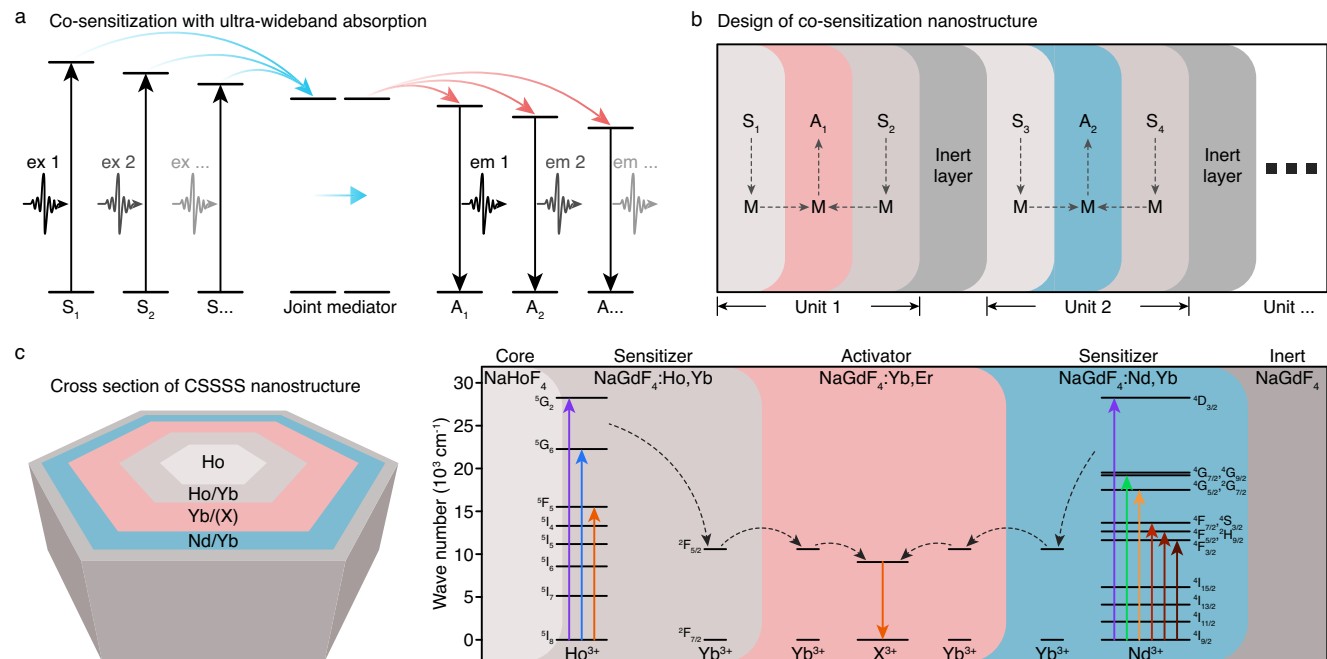

**Fig. 1 | Schematic illustration of co-sensitization strategy and design of symmetric multilayer core-shell structure. a** Mechanism of the ET within a co-sensitization structure (S: Sensitizer, A: Activator). **b** Illustration of a universal ultrawideband-responsive structure with multiple optical units (S: Sensitizer, M: Mediator, A: Activator). **c** Illustrated cross-section of prepared CSSSS with symmetric penta-layer core-shell structure (X = Er, Ho, Pr, and Tm).

respectively[21,22]. However, these pioneering works and other analogous strategies[23,24] still suffer from the narrowband absorption of a single sensitizer.

Here, we report a co-sensitization strategy to achieve ultrawideband absorption and fine manipulation of both upconversion and downshifting luminescence of LnNCs. This co-sensitization strategy was established by adopting multiple $Ln^{3+}$ as co-sensitizers, a $Ln^{3+}$ as the joint mediator, and a group of $Ln^{3+}$ as activators (Fig. 1a and Supplementary Fig. 1c), which we referred to as the *co-sensitizer-mediator-activator triplet* that constitutes a basic optical unit (Fig. 1b). The combination of two or more sensitizers endows an ultra-wideband absorption to LnNCs, and a joint mediator, instead of multi-mediators, is capable of transporting excitation energy from each sensitizer to activators. In contrast to the widely used asymmetric sequential core-shell structure (Supplementary Fig. 1b) in single sensitization, a symmetric core-shell structure was fabricated with two peripheral sensitizer/mediator layers surrounding the central activator/mediator layer (Supplementary Fig. 1c). As a result, the proposed co-sensitization strategy allows simultaneous visible and second near-infrared (NIR-II) fluorescence with an ultrawide response from UV to NIR, thus satisfying different excitation demands (e.g., low-powered white light) as opposed to the few choices of isolated high-energy lasers (e.g., 808 and 980-nm lasers). These features endow the LnNCs with compelling application outcomes in low-powered white-light-mediated bioimaging and excitation-regulated encryption.

## Results

### Principle of the ultra-wideband-responsive LnNCs

To verify the feasibility of the proposed co-sensitization strategy, we first explored the capability of $Ln^{3+}$ candidates as co-sensitizers. Among the five typical $Ln^{3+}$ of $Pr^{3+}$, $Nd^{3+}$, $Ho^{3+}$, $Er^{3+}$, and $Tm^{3+}$ that had multiple responsive bands (Supplementary Fig. 2) and characteristic NIR-II emissions, $Ho^{3+}$ and $Nd^{3+}$ were found to show efficient ET to $Yb^{3+}$ (Supplementary Fig. 3). This was attributed to the good match of energy levels of $Ho^{3+}$ and $Nd^{3+}$ with that of $Yb^{3+}$[25,26], as evidenced by the intense $Yb^{3+}$ emission of $NaGdF_4$:$Ho^{3+}$,$Yb^{3+}$ and $NaGdF_4$:$Nd^{3+}$,$Yb^{3+}$ under the excitations of $Ho^{3+}$ and $Nd^{3+}$, respectively (Supplementary

Fig. 4). The >4 times faster decay rates of $Ho^{3+}$ and $Nd^{3+}$ than that of $Yb^{3+}$ in a $NaGdF_4$ matrix (Supplementary Fig. 5) offered the favorable conditions for efficient ET. Besides, $Ho^{3+}$ and $Nd^{3+}$ were found to present maximally complementary absorption characteristics with abundant narrow bands spanning from 300 to 900 nm (Supplementary Fig. 2), thus promoting themselves as model co-sensitizers. Being an excellent energy-migratory mediator and a sensitizer[27,28], $Yb^{3+}$ was testified to be capable of efficiently harvesting the excitation energy absorbed by $Ho^{3+}$ and $Nd^{3+}$ and transferring to four NIR-II fluorescent activators ($Er^{3+}$, $Ho^{3+}$, $Pr^{3+}$, and $Tm^{3+}$) (Supplementary Fig. 6). The role of $Yb^{3+}$ as a mediator for ET between sensitizers and activators in core-shell structures was evaluated by using two model structures without or with $Yb^{3+}$ doping in the intermediate layer between a $Ho^{3+}$/$Nd^{3+}$-sensitization layer and an $Er^{3+}$-emission layer. The prominent enhancement in $Er^{3+}$ emissions was detected only with the presence of $Yb^{3+}$ (Supplementary Fig. 7). Collectively, these results demonstrated that $Yb^{3+}$ could be the pivotal joint mediator in the proposed *co-sensitizer-mediator-activator triplet* to efficiently manipulate energy transfer and photon conversions in rationally designed LnNCs via co-sensitization.

We designed co-sensitized LnNCs by using $Ho^{3+}$ and $Nd^{3+}$ as co-sensitizers, $Yb^{3+}$ as the joint mediator, and $Er^{3+}$ (or $Ho^{3+}$, $Pr^{3+}$, and $Tm^{3+}$) as the activators in a symmetric penta-layer core-shell (core/shell/shell/shell/shell, CSSSS) structure (Fig. 1c). In particular, the activator layer was surrounded by two spatially separated sensitizer layers to avoid unexpected ET between $Ho^{3+}$ and $Nd^{3+}$[29] (Supplementary Fig. 8), thus governing the directional ET from sensitizers to the mediator and then to the activator. $NaGdF_4$ was selected as the matrix since it has been demonstrated as one of the best hosts for $Ln^{3+}$[19,21,30]. The optimal molar doping amounts of two sensitizer/mediator layers to obtain excellent optical performances were determined to be 20%$Ho^{3+}$/1%$Yb^{3+}$ and 5%$Nd^{3+}$/5%$Yb^{3+}$, respectively (Supplementary Figs. 9 and 10 and Supplementary Table 1). Notably, a pure $NaHoF_4$ core was introduced to compensate for the relatively weaker absorption of $Ho^{3+}$ in comparison to $Nd^{3+}$ to tune the excitation spectrum (Supplementary Fig. 2), and the outermost inert shell of $NaGdF_4$ avoided the overflow of energy to peripheral quenchers (Fig. 1c).

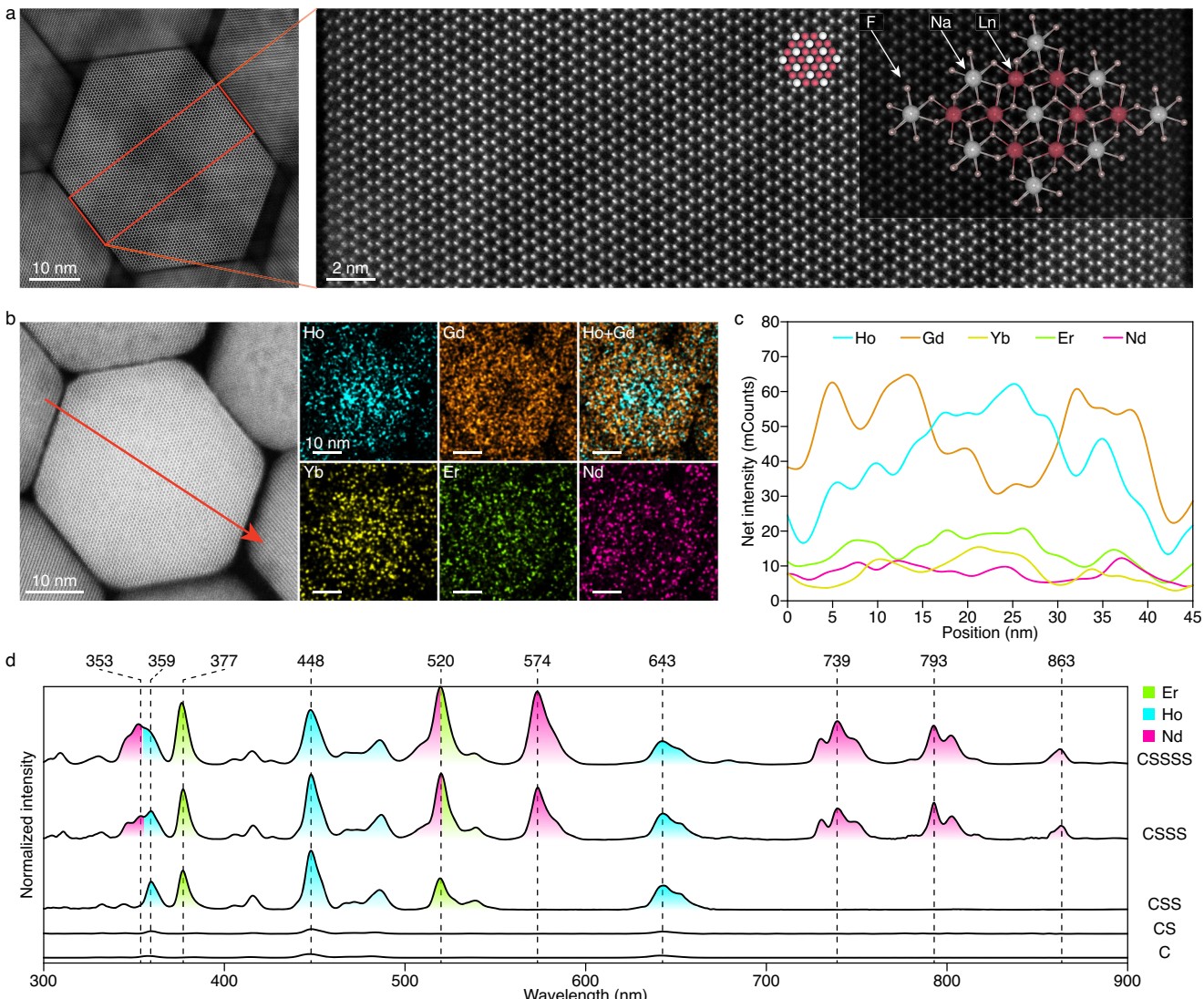

**Fig. 2 | Characterization of penta-layer core-shell NCs. a** High-resolution HAADF image of Er-NCs. Inset was the ball-and-stick model of hexagonal NaGdF$_4$. **b** STEM-EDS elemental mapping of Er-NCs. **c** Distributions of Ho, Gd, Yb, Er, and Nd determined by line scan showed in **b. d** Evolution of excitation spectrum (em = 1527 nm) of Er-NCs from C, CS, CSS, CSSS, to CSSSS.

The core-shell structured LnNCs were then synthesized sequentially by a hot injection method from the NaHoF$_4$ core. The coating schemes regarding temperature and the feed rate of precursors were explored to ensure the coherent epitaxial growth of shell-precursors on core NCs (Supplementary Fig. 11). These parameters were applicable for all coating processes, as tested by the preparation of CS and CSS LnNCs (Supplementary Fig. 12). The precursor amounts of Ho-layer and Nd-layer were critical to balance their respective contributions of absorption, and a molar ratio of Ho-layer/Nd-layer precursors of 1.0/1.5 presented analogous intensities of characteristic absorption bands that originated from individual Ho$^{3+}$ and Nd$^{3+}$ (Supplementary Figs. 13–15). The precursor amount of the activation layer (Er-layer in this case) was determined to be 1.5 times that of the Ho-layer to guarantee the overall excitation spectrum with similar intensity (Supplementary Figs. 16 and 17). Then through successive shell growth (Supplementary Fig. 18), we successfully prepared four kinds of LnNCs (Er/Ho/Pr/Tm-NCs) with a CSSSS structure by simple substitution of activator precursors (of Er$^{3+}$, Ho$^{3+}$, Pr$^{3+}$, and Tm$^{3+}$), which showed uniform hexagonal morphology with an average size of ~41 nm (Supplementary Fig. 19). The perfect atomic arrangement across the whole crystal demonstrated the high crystallinity of Er-NCs and the

robusticity of the regulation of shell growth (Fig. 2a). X-ray diffraction (XRD) result confirmed the hexagonal phase structure (Supplementary Fig. 20). EDS mapping, line scan analyses, and electron energy loss spectroscopy (EELS) analyses consistently revealed the clustered distributions of lanthanide ions (especially the high-amount Gd, Ho, and Er) in different regions of a single nanocrystal (Fig. 2b, c and Supplementary Fig. 21), which provided rational evidence to support the designed penta-layer core-shell structure (Fig. 1c)[21]. The excitation spectrum was monitored with a gradually increased number of excitation bands from C to CS, CSS, CSSS, and CSSSS (Fig. 2d). An ultrawide excitation with up to nine characteristic narrow bands was finally obtained, covering almost the full wavelength region from 350 to 850 nm.

**Ultra-wideband-responsive photon conversion of LnNCs**

Under the excitation at discrete wavelengths of 353, 377, 448, 520, 574, 643, 739, 793, 863 nm, as well as the conventional 980 nm, a broad range of emissions of Er-NCs from visible to NIR-II were observed (Supplementary Fig. 22). The three characteristic peaks at 540, 654, and 1527 nm corresponded to the $^4S_{3/2}$-$^4I_{15/2}$, $^4F_{9/2}$-$^4I_{15/2}$, and $^4I_{13/2}$-$^4I_{15/2}$ transitions of Er$^{3+}$, respectively. Other signals detected under partial

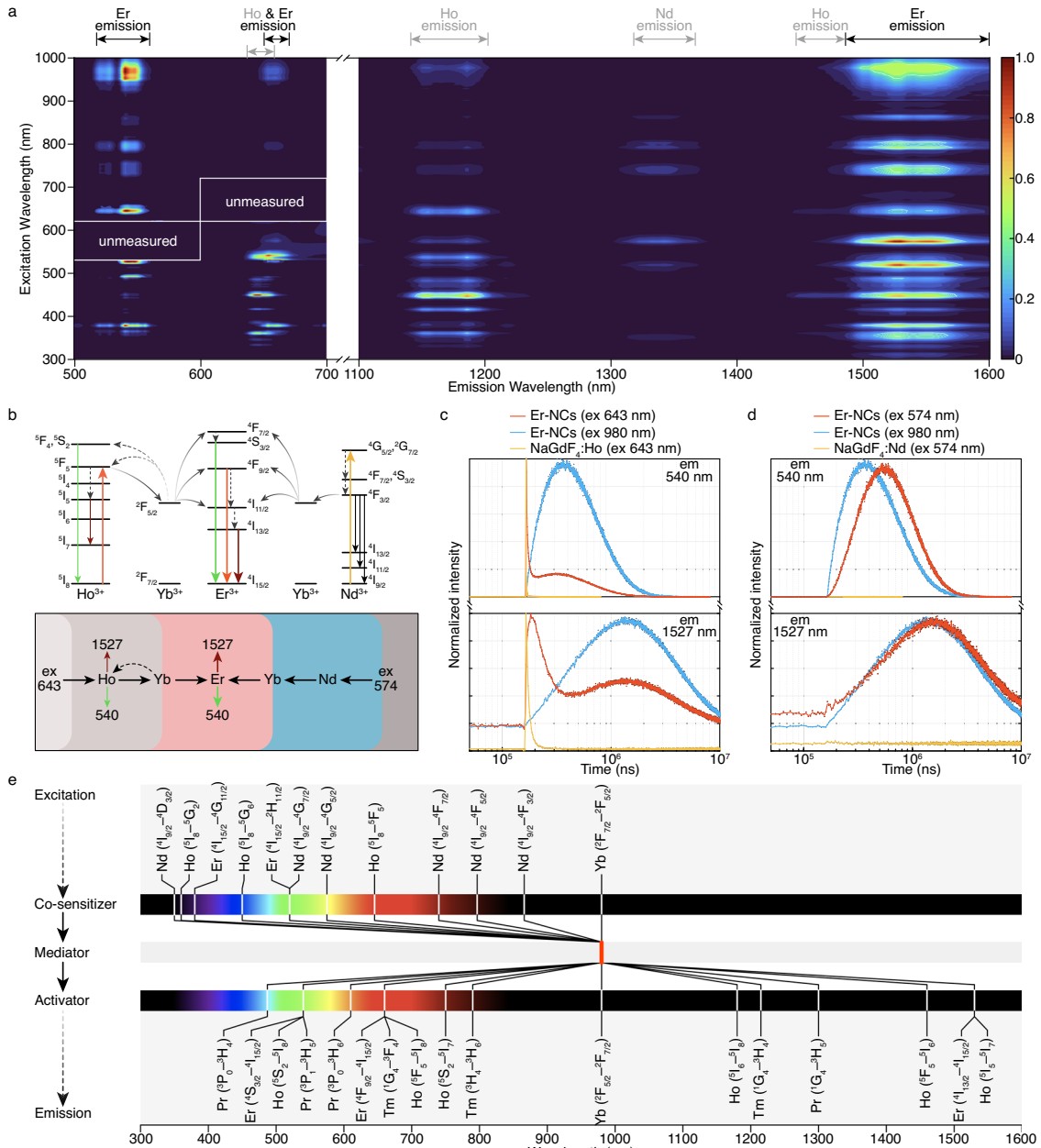

**Fig. 3 | Ultra-wideband-responsive photon conversion of LnNCs. a** Excitation-emission map of Er-NCs. Data within two marked areas were hard to measure as the emission band was overlapped with the excitation band. **b** ET paths of Er-NCs under the excitations of 643 and 574 nm (emission of $Yb^{3+}$ at 980 nm was left out for clarity). **c** Transient spectra of Er-NCs under the excitation of 643 and 980 nm and of NaGdF$_4$:Ho under 643 nm excitation. **d** Transient spectra of Er-NCs under the excitation of 574 and 980 nm and of NaGdF$_4$:Nd under 574 nm excitation. **e** A summary of the major absorptions and emissions in the *co-sensitizer-mediator-activator triplet* with corresponding transitions annotated.

excitation conditions were indexed to the intrinsic emissions of $Ho^{3+}$ (at 1180 nm and 1460 nm) and $Nd^{3+}$ (at 1340 nm). The possible transitions of energy levels corresponding to major absorptions and the visible/NIR emissions of Er-NCs were proposed in Supplementary Fig. 23. Further excitation-emission map indicated that the ultra-wideband excitation in the range of 350-1000 nm could spur intense emissions of Er-NCs at both visible and NIR-II regions, intuitively presenting a many-to-many relationship between excitation (multiple bands from 350-1000 nm) and emission (at 540, 654, and 1527 nm) (Fig. 3a) with temperature-dependent optical stability (Supplementary Fig. 24). This also confirmed that the co-sensitization strategy endowed Er-NCs with abundant choices of excitation beyond that of a single sensitization case. The strategy to achieve ultra-wideband-responsive photon conversion at both visible and NIR-II regions was also applied to Ho-, Pr-, and Tm-NCs (Supplementary Figs. 25 and 26). The photon conversion processes from co-sensitizers to the activator through the bridge of the joint mediator were then investigated by transient spectra. The typical green emission at 540 nm and NIR-II emission at 1527 nm of Er-NCs were selected as representative processes for analysis. In response to various excitations, the transient spectra at both emissions presented two modes, including a rise-decay mode and a rise-decay-rise-decay mode (Supplementary Figs. 27 and 28). The difference in the profiles of transient spectra implied an excitation-dependent ET behavior in Er-NCs.

Therefore, the analysis of transient processes for $Ho^{3+}$-$Yb^{3+}$-$Er^{3+}$ and $Nd^{3+}$-$Yb^{3+}$-$Er^{3+}$ was conducted by picking the characteristic

absorption of Ho$^{3+}$ at 643 nm and of Nd$^{3+}$ at 574 nm, respectively (Fig. 3b). For 643 nm excitation, the Ho$^{3+}$ in Er-NCs absorbed the excitation energy. One part of the excitation energy was depleted by Ho$^{3+}$ itself through the radiation transitions of $^5S_2$-$^5I_8$ (emission at 540 nm) and $^5I_5$-$^5I_7$ (emission at 1527 nm) (Supplementary Fig. 29), as evidenced by the transient spectra of the control sample of NaGdF$_4$:Ho$^{3+}$ (Fig. 3c). While another part of the excitation energy was transferred to Er$^{3+}$, followed by the radiation transitions of Er$^{3+}$ of $^4S_{3/2}$-$^4I_{15/2}$ (emission at 540 nm) and $^4I_{13/2}$-$^4I_{15/2}$ (emission at 1527 nm). Thus, the monitoring of the transient responses of Er$^{3+}$ at 540 and 1527 nm would mix with the signal from Ho$^{3+}$, i.e., the rise-decay-rise-decay mode of the transient spectrum was a superposition of two transient processes with different decay rates (Fig. 3c). Notably, the transient spectrum of Er-NCs with 980 nm excitation presented a rise-decay mode, which was attributed to the single step energy transfer from Yb$^{3+}$ to Er$^{3+}$. For 574 nm excitation, the Nd$^{3+}$ in Er-NCs absorbed the excitation energy but did not produce any interfering emission with that of Er$^{3+}$ at either 540 or 1527 nm. The energy was transferred to Yb$^{3+}$ and then to Er$^{3+}$ step by step, followed by the radiation transitions of Er$^{3+}$ at 540 and 1527 nm. So, the transient spectrum presented a conventional rise-decay mode (Fig. 3d). These two different modes of transient spectra visualized distinct ET paths of Er-NCs in response to different excitations of Ho$^{3+}$ and Nd$^{3+}$ and affirmed the capacity of the co-sensitized symmetric core-shell structure in controllable manipulation of directional ET and multidirectional photon conversion, which were found to be superior to that of NCs with elements co-doping in one matrix (Supplementary Fig. 30) or with asymmetric or isolated sensitized structures (Supplementary Figs. 31 and 32). As a result, the combination of co-sensitization and ET manipulation collectively contributed to the ultra-wideband-responsive multidirectional photon conversion of lanthanide nanocrystals (Fig. 3e).

### White light-excited whole-body angiography of mice

The feature of ultra-wideband excitation and intense NIR-II emission at 1527 nm indicated the prospect of Er-NCs as a low-energy-excitable NIR-II fluorescent probe for real-time high-resolution bioimaging. A white light-emitting diode (WLED) and five monochromic LEDs centered at 450, 520, 570, 740, and 800 nm, corresponding to the excitation peaks of Er-NCs at 448, 520, 574, 739, and 793 nm (Supplementary Fig. 33), were employed to evaluate the performance of Er-NCs that placed in a lab-made imaging system with LEDs, an InGaAs camera, and a long-pass filter at 1500 nm (Fig. 4a). Upon exposure to each monochromic LED for 2 ms, obvious NIR-II fluorescence of Er-NCs was captured under the excitation of the visible LEDs at 450, 520, and 570 nm, as well as the conventional NIR-LEDs at 740 and 800 nm (Fig. 4b). The fluorescence of Er-NCs exhibited a positive correlation with the increase in the applied monochromic LEDs (Fig. 4c and Supplementary Video 1) due to the abundant excitation bands of Er-NCs achieved by co-sensitization. Further increase of exposure time to 25 ms induced intense NIR-II fluorescence, and a single WLED was found to be capable of exciting Er-NCs with comparable intense fluorescence (Supplementary Fig. 34). The low-energy-LED excitability of Er-NCs might be attributed to the efficient NIR quantum cutting of Ho$^{3+}$-Yb$^{3+}$ pair[31] and Nd$^{3+}$-Yb$^{3+}$ pair[32] in the sensitizer layers of our nanostructure (Supplementary Fig. 35), while the downshifting luminescence at 1527 nm might be regarded as a long-range NIR quantum cutting of Ho$^{3+}$/Nd$^{3+}$ to Er$^{3+}$ via the bridge of Yb$^{3+}$. Collectively, our co-sensitization strategy offered LnNCs bioimaging capability under the excitation by a low-energy white light source other than the specifically designed high-power lasers.

The oleic acid-capped Er-NCs were surface-modified with an amphiphilic polymer of poly(maleic anhydride-alt-1-octadecene)-PEG (PP) before in vivo bioimaging (Supplementary Figs. 36 and 37). The produced monodispersed Er-PP NCs exhibited good stability in normal saline solution (Supplementary Fig. 38) without apparent cytotoxicity (Supplementary Fig. 39). Upon excitation by an 808-nm laser and a WLED, the Er-PP NCs in normal saline solution exhibited strong NIR-II fluorescence at a concentration of 1 mg/mL (Fig. 4d), indicating the competence of a common white light source for bioimaging. Real-time whole-body angiography of mice with intravenously injected Er-PP NCs was first performed using an 808-nm laser (power density: 0.12 W cm$^{-2}$) and a WLED (power density: ~0.12 W cm$^{-2}$) as the excitation source, respectively (Fig. 4e). Both the 808-nm laser and the WLED excitations visualized main vascular structures (Fig. 4f,g and Supplementary Videos 2 and 3). The 808-nm laser excitation presented a higher resolution due to the higher tissue penetration of NIR light. The common WLED excitation enabled a comparable signal-to-noise ratio (SNR) of 12.3 with that of 808-nm laser excitation (16.1) in sketching the profiles of main vessels despite a moderate spatial resolution. A higher SNR of 19.0 was also obtained with a 980-nm laser (Supplementary Fig. 40 and Supplementary Video 4), which might be attributed to the strong absorbance at 980 nm by the mediator of Yb in Er-NCs. Due to the high absorptivity of Er-NCs to WLED (0.3515 m$^2$), 808-nm laser (0.2687 m$^2$), and 980-nm laser (0.9867 m$^2$) (Supplementary Fig. 41), the excitation sources of LnNCs for high-resolution bioimaging could therefore be extended to visible light beyond previously reported sole NIR lasers or low-powered LED[3,33]. It is believed that the Er-PP probe with WLED-excitability might also inspire advanced and feasible NIR fluorescence-guided surgery in an open environment, thus eliminating inconvenience due to laser excitation (Fig. 4e).

### WLED-responsive multiplex encryption of LnNCs

The feature of ultra-wideband response of LnNCs also inspired excitation-regulated applications, like multiplex encryption and decryption. The regulation of excitation behaviors of LnNCs was realized by using a photochromic film composed of a photo-acid generator (PAG) and a pH-sensitive dye (bromocresol green, BG). The released acids from PAG upon exposure to a handheld UV lamp (365 nm) would induce the pH-dependent color (absorption) change of BG (Fig. 5a and Supplementary Figs. 42 and 43), endowing the photochromic film with a complete block region between 300-450 nm, a tunable transmit region between 450-700 nm, and a total transparent region beyond 700 nm (Fig. 5b and Supplementary Video 5). The tunable optical transmittance was closely correlated with the UV exposure time. In particular, the film with no exposure only allowed <35% transmittance below 630 nm and a 4-min exposure provided a maximal transmittance (>35% between 470 to 530 nm, and 100% beyond 530 nm). As a result, the film enabled fine control over the available excitation bands and intensities of Ho-NCs (Em: 1186 nm) and Er-NCs (Em: 1527 nm). Due to the maximal transmittance of the film, the accessible 450-700 nm of a WLED could excite encrypted LnNCs-printed patterns with clear visualization of a quick response code and ciphertext of "ILXE" (corresponding to the plaintext of "ROAD") (Fig. 5c and Supplementary Fig. 44). Moreover, the time-dependent optical transmittance behavior realized a more complex encryption approach. As shown in Fig. 5d, the correct crane pattern could only be read out with a 1-min exposure and the presence of a long-pass filter at 1500 nm, while other conditions would present misleading results, which demonstrated the advantages of co-sensitized LnNCs via excitation-regulation.

### Discussion

In summary, we have demonstrated an efficient co-sensitization strategy to achieve ultra-wideband absorption of LnNCs. A rationally designed symmetric multilayer core-shell nanostructure governs the directional energy transfer from multiple sensitizers to activators,

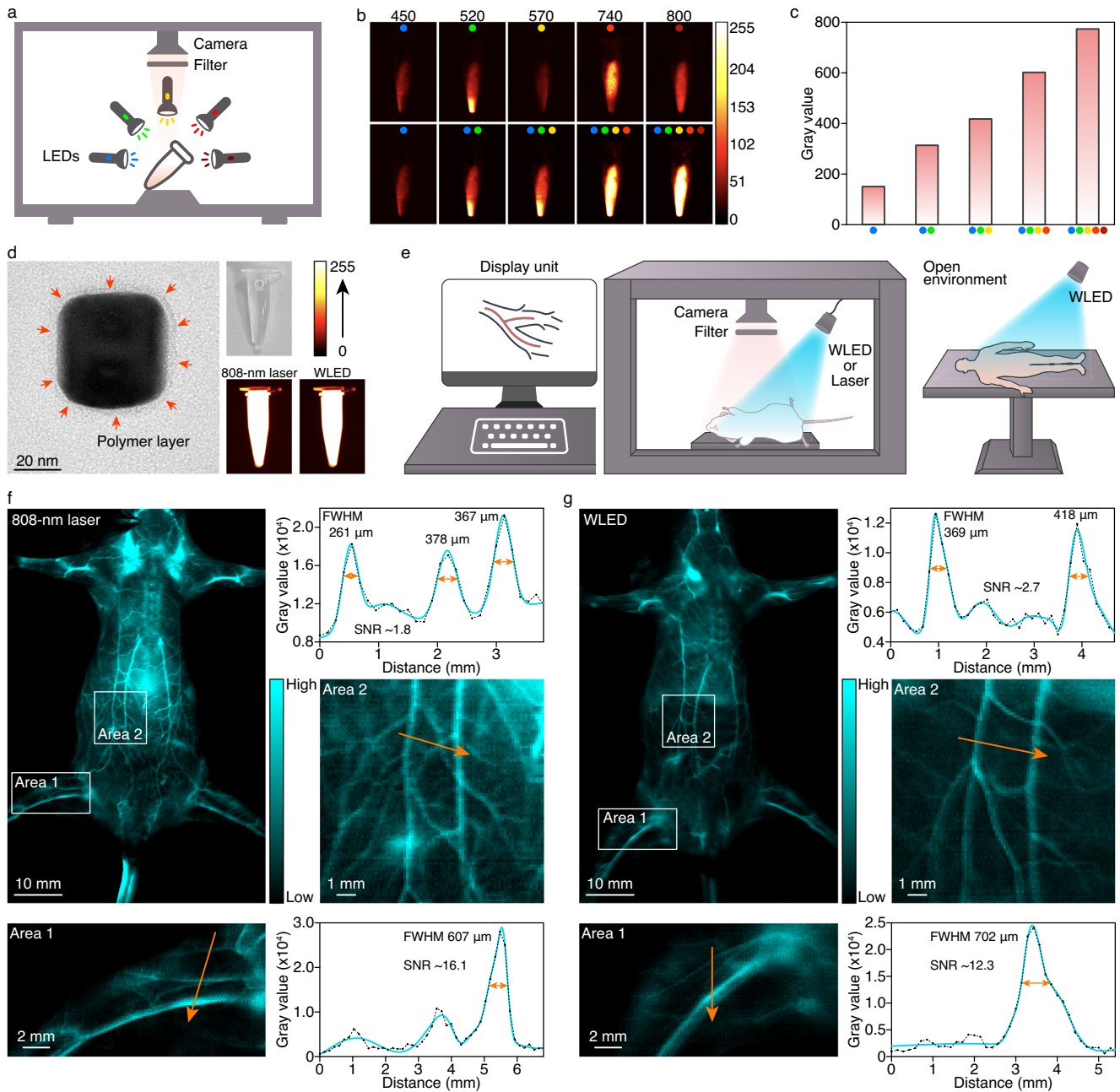

**Fig. 4 | White light-excited fluorescence of Er-NCs and whole-body angiography of mice. a** Illustration of the experimental setup for NIR-imaging. **b** NIR-imaging of Er-NCs exposed to single or multiple LEDs (exposure: 2 ms; filter: 1500LP). **c** Analysis of the gray values in **b**. **d** TEM image and NIR-imaging of Er-PP NCs in normal saline solution. **e** Illustration of whole-body angiography of mice (left) and envisaged WLED-excited diagnosis of human (right). **f** Whole-body angiography of a mouse using an 808-nm laser as the excitation source, and the analysis on cross-sectional intensity (black line) and Gaussian fit fluorescence intensity profiles (cyan line), FWHM: Full width at half maxima, SNR: Signal-to-noise ratio. **g** Whole-body angiography of a mouse using a WLED as the excitation source, and the analysis on cross-sectional intensity (black line) and Gaussian fit fluorescence intensity profiles (cyan line).

achieving a distinctive multidirectional photon conversion mode surpassing the excitation limits. Note that the intrinsic spontaneous radiation of sensitizers and a mediator may preclude the maximal conversion of excitation energy, further structural optimization of LnNCs to regulate the energy flux may help improve the efficiency and promote functional diversification. The long-range downshifting NIR-II fluorescence endows the LnNCs with excellent performance in low-powered WLED-mediated angiography and excitation-regulated multiplex encryption. Moreover, the ultra-wideband responsive characteristics originated from our co-sensitization strategy also open possible avenues for chemiluminescence- or bioluminescence-excited NIR in vivo bioimaging applications and NIR fluorescence-guided

surgery using common white-light as illumination sources (Fig. 4e). Overall, this work presents a new way of tuning photonic behaviors of lanthanide nanocrystals through a *co-sensitizer-mediator-activator triplet*, which may stimulate advanced materials design and applications.

## Methods
### Materials
Holmium oxide (99.99%), gadolinium oxide (99.99%), ytterbium(III) oxide (99.99%), erbium oxide (99.99%), neodymium oxide (99.99%), thulium(III) oxide (99.99%), praseodymium(III, IV) oxide (99.99%), trifluoroacetic acid (TFA, 99.5%), sodium trifluoroacetate (Na(TFA), 97%), poly (methyl methacrylate) (PMMA), 2-(4-Methoxystyryl)−4,6-

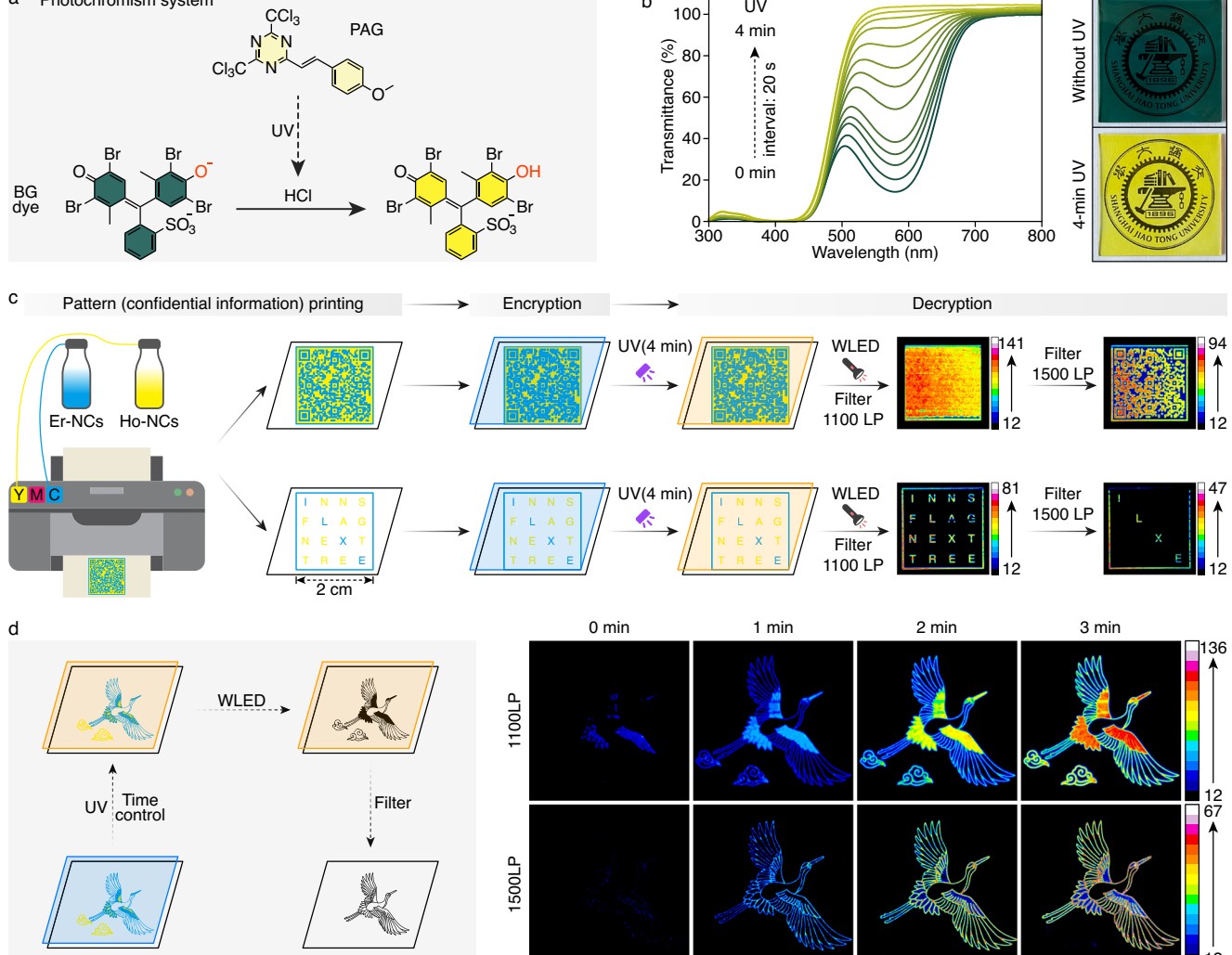

**Fig. 5 | WLED-responsive multiplex encryption and decryption. a** Photochromic system comprising a pH-sensitive dye (BG) and a PAG. **b** Transmittance curves of the photochromic film with different periods of UV exposure and images of the photochromic films before and after 4-min UV exposure. **c** Encryption and decryption of a quick response code and the ciphertext of "ILXE". **d** UV exposure time-controlled encryption and decryption of a crane pattern.

bis(trichloromethyl)−1,3,5-triazine (PAG, 98%), triethylamine (TEA, 99.5%), bromocresol green sodium salt (BG), nitrosonium tetrafluoroborate (NOBF₄, 98%) were purchased from *Macklin*. Bis(2-ethylhexyl) phthalate (DEHP, 95%) was purchased from *Bide Pharmatech Co., Ltd.* Oleic acid (OA, 90%), 1-octadecene (ODE, 90%), and poly(maleic anhydride-*alt*−1-octadecene) (PMH) were purchased from *Sigma-Aldrich*. mPEG-NH₂ (MW: 4 kDa), 4-dimethylaminopyridine (DMAP) were purchased from *Aladdin*. All of these materials were used as received unless otherwise noted.

### Preparation of lanthanide trifluoroacetate precursors
Five grams of lanthanide oxide (Ln₂O₃, Ln = Ho, Gd, Yb, Er, Nd, Tm, or Pr) were added to a 50-mL flask. To the flask was added excess water and trifluoroacetic acid (1:1). The mixture was then stirred at 100 °C under reflux until the solution became completely clear and transparent. After cooling to room temperature, this solution was filtered. Finally, the solvents of water and trifluoroacetic acid were evaporated under reduced pressure. A dry powder of lanthanide trifluoroacetate (Ln(TFA)₃) could be obtained with a yield of >90%.

### Synthesis of CSSSS NCs
(1) Synthesis of NaHoF₄ core NCs. The synthesis of core NCs was based on reported procedures[34]. To a 50-mL three-neck flask was added

Na(TFA) (2.0 mmol, 272 mg), Ho(TFA)₃ (1.0 mmol, 504 mg), OA (10 mL), and ODE (10 mL). The mixture was heated to 160 °C under N₂ flow to get a clear solution, followed by an evacuation (30 min) at 110 °C. Then the temperature was quickly increased to 330 °C under N₂ flow and maintained at that temperature for 15 min. Then the solution was cooled to room temperature and purified three times with ethanol and cyclohexane. The final product was dispersed in 3 mL of cyclohexane.

(2) Synthesis of NaHoF₄@NaGdF₄:20%Ho,1%Yb CS NCs. (A) The shell precursor was prepared first. To a 50-mL flask was added Na(TFA) (1.8 mmol, 245 mg), Gd(TFA)₃ (0.79 mmol, 392 mg), Ho(TFA)₃ (0.2 mmol, 101 mg), Yb(TFA)₃ (0.01 mmol, 5 mg), OA (4 mL), and ODE (6 mL). The solution was evacuated at 110 °C for 30 min and then cooled to room temperature under N₂ flow. (B) To a 50-mL flask was added NaHoF₄ NCs (1 mL), OA (8 mL), and ODE (8 mL). The solution was evacuated at 110 °C for 30 min. Then the temperature was increased to 280 °C under N₂ flow. The shell precursor was then added by a syringe pump at a speed of 6.5 mL h⁻¹. After that, the solution was maintained at that temperature for 30 min and then cooled to room temperature. The product was purified three times with ethanol and cyclohexane.

(3) Synthesis of multilayer core-shell NCs. The synthesis of multilayer core-shell NCs followed the same procedures as that of CS NCs,

except that the compositions and amounts need to be changed accordingly (Supplementary Table 1).

## Characterizations

Absorption spectra were measured using a UV/VIS/NIR Spectrophotometer (Lamda 950, PerkinElmer). TEM images, compositional mapping, and Selected Area Electron Diffraction (SAED) were captured/conducted using a Scanning/Transmission Electron Microscope (STEM, Talos F200X G2). X-ray diffraction was performed using a mini-X-Ray diffractometer with Cu Kα radiation (λ = 1.5418 Å, Japan). The Fourier transform infrared (FTIR) spectra were obtained by the FTIR spectrometer (Nicolet 6700, Thermo Scientific).

## Optical characterizations

(1) NIR excitation and emission spectra were measured by a photoluminescence spectrometer of FLS-1000 (Edinburgh Instruments) using a Xe-lamp as the excitation source.

(2) For visible emission spectra measurement, the samples were excited by a femtosecond optical parametric amplifier (TOPAS-F-UV2, Spectra-Physics) with a tuning range from 240 to 2600 nm.

(3) For the measurement of the excitation-emission map, the excitation sources of a Xenon lamp and a femtosecond optical parametric amplifier (fs OPA) and the filters of 510 LP, 530 LP, 626 LP, 575 SP, 690 SP, and 1000 LP were adopted. For each sub-map, the adopted configuration of the excitation source and filter was marked in Supplementary Fig. 45. The maximum intensity of each sub-map was normalized to 1.

(4) For time-resolved photoluminescence measurement, a tunable mid-band optical parametric oscillator (OPO) pulse laser was used as the excitation source (410–2400 nm, 10 Hz, pulse width ≤5 ns, Vibrant 355II, OPOTEK). Data were acquired by a multi-channel scanning unit.

(5) For temperature-dependent photoluminescence measurements, a low-temperature accessory (OptistatDN, Oxford Instruments) and a high-temperature accessory (T95-PE, Linkam Scientific) were used as the temperature controller. The sample was maintained at a specific temperature for 10 min, then the emission spectrum was collected.

## Synthesis of Er-PP probe

(1) Synthesis of the amphiphilic polymer, PMH-PEG (PP). To a 20-mL vial was added mPEG-NH$_2$ (85 mg) and THF (7 mL), followed by 5 min of ultrasonication to get a clear solution. Then to the PEG solution was added 1 mL of PMH (in tetrahydrofuran, 10 mg mL$^{-1}$), DMAP (4 mg), and TEA (6 μL). The solution was stirred overnight under dark conditions.

(2) Surface modification of Er-NCs by PP. The as-prepared PP solution was mixed with Er-NCs (20 mg in 2 mL of chloroform) and allowed for 4 h of agitation. Then the solvent was evaporated under reduced pressure to get a light-yellow powder. The powder was dispersed in deionized water to get a clear solution, followed by dialysis against DI water for 2 days to remove any impurities. The purified solution was then freeze-dried to get the pure Er-PP powder. The purified Er-PP NCs could be redispersed in normal saline upon 5 min of ultrasonication, getting a clear solution.

## In vitro cytotoxicity assay

4T1 cells were cultured in RPMI 1640 media with 10% FBS and 1% penicillin/streptomycin. Cells were incubated at 37 °C in a humidified atmosphere with 5% CO$_2$. 4T1 cells were seeded in a 96-well plate (10$^4$ cells well$^{-1}$) for 12 h. The media was then replaced with fresh media containing different concentrations of Er-PP NCs (0–1000 μg mL$^{-1}$) and allowed for 24 h of incubation. The medium was then discarded, and cells were washed with PBS three times. Cell viabilities were

determined using a CCK-8 kit. Each concentration was performed with six repetitions.

## Animal experiments

Female Balb/c mice (8 weeks) were purchased and maintained under protocols approved by the Shanghai Jiao Tong University Laboratory Animal Center. Before imaging, mice were anesthetized with 2 L min$^{-1}$ air mixed with 4% isoflurane. Then the hairs on the abdomen and legs of mice were removed by depilated cream.

## NIR-II fluorescence imaging of Er-NCs and whole-body angiography of mice

(1) For the imaging of the powder Er-NCs sample, the sample was excited by several monochrome LEDs with central wavelengths at 450, 520, 570, 740, and 800 nm, respectively. A white LED was also adopted. The rated power of all LEDs was 5 W. The images were captured by an InGaAs camera (Artemis Intelligent Imaging, Shanghai, China) with an exposure of 25 ms. Then the exposure was reduced to 2 ms, and multiple LEDs were turned on and off gradually. The images and video were recorded in the process.

(2) For the whole-body angiography, mice were anesthetized with 2 L min$^{-1}$ air mixed with 4% isoflurane throughout the imaging process. An 808-nm laser (power density: 0.12 W cm$^{-2}$), 980-nm laser (power density: 0.12 W cm$^{-2}$), and a WLED (power density: -0.12 W cm$^{-2}$) were adopted as the excitation sources to compare their imaging effects. The whole process from the injection of the probe to the visualization of vessel structures was recorded by using the NIR camera with a 1500 nm LP filter (Thorlabs), and mice were intravenously injected with the Er-PP probe by a drainage needle with a total amount of ca. 50 mg kg$^{-1}$. The acquisitions lasted for several minutes. Data were processed using ImageJ software.

## Preparation of photochromic film

To a 20-mL vial was added PMMA (2.25 g) and acetone (15 mL). The mixture was stirred for 2 h to get a transparent solution. To this solution was added BG dye (2.0 mg), DEHP (0.4 mL), PAG (1.0 mg), and TEA (2 μL). After stirring for 30 min, the solution was poured into a glass Petri dish (Φ = 60 mm) and dried at 30 °C overnight (higher temperature may cause the production of bubbles). Finally, the film was cut into a 4 × 4 cm square and preserved in a dark place for further use.

## Information encryption and decryption application

(1) Preparation of security ink. (A) Ligand exchange with NOBF$_4$. Typically, a total of 50 mg Er-NCs (or Ho-NCs) were dispersed in 2 mL of cyclohexane followed by the addition of 5 mL of NOBF$_4$ (10 mg mL$^{-1}$ in DMF). The NCs were transferred to the DMF phase after ultrasonication for 5 min. The product was washed twice with ethanol to remove any remaining nonpolar organics. (B) Er-ink. The purified Er-NCs were dispersed in a 5-mL mixed solvent containing ultrapure water (20%), ethanol (50%), and glycerol (30%). (C) Ho-ink. The purified Ho-NCs were dispersed in a 2.5-mL mixed solvent containing ultrapure water (20%), ethanol (50%), and glycerol (30%).

(2) Printing of encrypted pattern. The printing of the encrypted pattern was performed using a HP DeskJet 1112 printer. First, the original tri-color ink cartridge was washed thoroughly using deionized water and ethanol and dried in a vacuum oven. Then the as-prepared Er-ink was injected into the grid original placing C-ink, while the Ho-ink was injected into the grid original placing Y-ink. All patterns were printed on commercially available color-inkjet paper.

(3) Information encryption/decryption application. For Information encryption, the pattern carrying confidential information was covered by the photochromic film (BG film), forming a security device (termed PBG). For Information decryption, the PBG was first irradiated by a handheld UV Lamp (365 nm), triggering the gradual photochromism of BG film from green to yellow. In this process, the

transmittance of this film to white light increased gradually. A sequence of pseudocolor images could be captured by a NIR camera. With the right filter, the real information could be read out. Similarly, this technology could also provide encryption and decryption for the QR code and the ciphertext of Hill cipher.

## Data availability

The data that support the findings of this work are available from the corresponding authors upon reasonable request. Source data are provided with this paper.

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

## Acknowledgements

This work was financially supported by National Natural Science Foundation of China, Project No. 81971704 (W.L.), the National Key Research and Development Program of China, Project No. 2017YFA0205304 (W.L.), and Translational Medicine Research Fund of National Facility for Translational Medicine (Shanghai), Project No. TMSK-2021-117 (W.L.). We thank Instrumental Analysis Center of SJTU for the assistance with optical and TEM characterizations. We also thank Zhejiang Orient Gene Biotech Co., Ltd. for their support.

## Author contributions

W.L. supervised and directed the study. Z.J., X.Y. and W.L. conceived the idea and designed the experiments. Z.J. performed the synthesis and characterization of LnNCs, conducted the whole-body angiography of mice and multiplex encryption and decryption with help from L.H. and Z.Y. The manuscript was written by Z.J., X.Y., and W.L. H.Q. and X.C. helped polish the manuscript. All authors participated in the discussion and analysis of experimental results and manuscript.

## Competing interests

The authors declare no competing interests.
