## [Peer review file · Nature Communications]

REVIEWER COMMENTS

Reviewer #1 (Remarks to the Author):

The paper demonstrated a co-sensitization strategy to achieve ultra-wideband absorption of LnNCs, by synthesizing a symmetric multilayer co-sensitizer core-shell nanostructure. This strategy and the results are interesting, but more details should be needed for the better understanding of the results.

1. As the authors claimed, formation of the core-shell nanostructure is critical for the ultra-wideband absorption property of the material. How was the core-shell nanostructure derived from the microstructure characterizations? It seems to me difficult to see a clear CSSSS (as illustrated in Fig. 1) simply from the EDS shown in Fig. 2. Because it is usually considered that the manipulation of elemental distribution at the atomic scale is difficult, evidences for more other CSSSS should be also provided to confirm the core-shell structure.
2. How can it be evidenced for the critical role of the shell-core nanostructure in the photon conversion properties, as compared with that of the simply combination of various elements? How about the material with same elemental doping but without forming the special core-shell nanostructure? Some more discussions should be needed.
3. Besides the doping ratio, what are the possible effects of the shape, size and impurities on the photon conversion of different CSSSS? And what is the possible effect of the powder size on the elemental distribution (formation of CSSSS)
4. The thermal stability of the CSSSS materials should be evaluated and discussed for the photon conversion application.

Reviewer #2 (Remarks to the Author):

The manuscript by Zhao et al. reports a type of co-sensitized LnNCs for WLEDs-mediated whole-body angiography of mice and excitation-regulated encryption. The authors have performed many control experiments, which make their experimental findings of robust low-powered WLED pumped LnNCs-based emission very convincing. The result is very interesting and updates the progress in the application of lanthanide materials excited by ultralow-cost light sources. However, I still have some concerns about the novelty or significance of this work. I expect something more than one application.

There are some specific comments and suggestions:

1. On page 6, Fig. 3. It maybe not be clear enough for the transient spectrum of NaGdF₄:Ho (ex643 nm). Actually, it exhibits a fast rise-decay profile. However, it seems to be a straight line in Fig. 3c. In addition, the recombination processes from the Nd³⁺→Nd³⁺ and Ho³⁺→Ho³⁺ transitions should be presented in Fig. 3b in terms of solid arrows.

2. On page 22, Supplementary Fig. 27b. According to the description in the main text, the emission spectra of NaGdF₄:Ho which is excited by the monochromic LED at the wavelength of 448 nm should be changed to that being excited by 643-nm LED.

3. Regarding the whole-body angiography experiment, NaHoF₄@NaGdF₄:Ho/Yb@NaGdF₄:Yb/Er@NaGdF₄:Nd/Yb@NaGdF₄ was utilized as the sample agent for its intense NIR-II emission at 1527 nm (Er³⁺, 4I13/2-4I15/2). In the main text, the authors have performed real-time whole-body angiography of mice under the excitation of 808-nm laser and a WLED, respectively. It is verified that the performance of Er-NCs excited by WLED is superior to that of the 808-nm laser pumped case (SFig.37). It is not reasonable for this experimental settings. Since the 808-nm energy can only be absorb by Nd³⁺ ions, and then transferred to Er³⁺ by Yb-sublattice, while there are no response from the NaHoF₄@NaGdF₄:Ho/ Yb@NaGdF₄:Yb/Er part at all under the excitation of 808-nm laser. Hence, the author should conduct the comparative tests for Er-NCs under the excitation of a WLED and 980-nm laser (instead of 808-nm one), particularly the adsorptivity data of WLED and 980-nm laser. Note that, the heating effect can be ignored when the pumping density of 980-nm laser diode is clamped at ~0.12 W cm⁻².

4. The absorption spectra of Ln³⁺-NCs (Ln = Er, Tm, Pr, Ho) are quite important to assess their performances. However, i cannot find those data until the calculated absorption cross-section of Er-NCs in Supplementary Fig. 37.

Reviewer #3 (Remarks to the Author):

This manuscript describes a heterogenous design of rare earth doped nanoparticles that can absorb the broad band LED light and emit both upconversion in visible and down-shifting luminescence in near infrared second window for animal imaging and anticounterfeiting (data storage) applications. The investigation is comprehensive with a vast amount of synthesis and systematic characterizations of the materials as well as two directions of application demonstrations. Most of the data here are convincing and coherent. The strategy of co-sensitization by employing Nd³⁺ and Ho³⁺ as the co sensitizers with

complementary absorptions from the ultraviolet to infrared are convincing. The formula of $20\%Ho^{3+}/1\%Yb^{3+}$ and $5\%Nd^{3+}/5\%Yb^{3+}$, designed in the penta-layer core-shell (core/shell/shell/shell/shell, CSSSS) structure can provide an excellent reference for the community to repeat the experiments. Remarkably, the low-powered white LED illumination suggests the potentials in the whole-body angiography of mice. I do not have other major comments except that the illustration of Figure 1 b was not consistent to the actual design of the nanoparticles, which I suggest this part of Figure should be re-drawn.

Author Rebuttals to Initial Comments:

Reviewer #1:

The paper demonstrated a co-sensitization strategy to achieve ultra-wideband absorption of LnNCs, by synthesizing a symmetric multilayer co-sensitizer core-shell nanostructure. This strategy and the results are interesting, but more details should be needed for the better understanding of the results.

Response: We thank the reviewer for the positive comment of our work as “interesting”, and for presenting several important points where we can improve our manuscript.

Comment 1. As the authors claimed, the formation of the core-shell nanostructure is critical for the ultra-wideband absorption property of the material. How was the core-shell nanostructure derived from the microstructure characterizations? It seems to me difficult to see a clear CSSSS (as illustrated in Fig. 1) simply from the EDS shown in Fig. 2. Because it is usually considered that the manipulation of elemental distribution at the atomic scale is difficult, evidence for more other CSSSS should be also provided to confirm the core-shell structure.

Response: We agree with the reviewer on this concern. Besides the presented TEM images, excitation spectra of Er-NCs, and STEM-EDS results, we have also conducted electron energy loss spectroscopy (EELS) analysis and photon conversion performance comparison between Er-NCs and three other types of NCs. While we admit that these results can not help confirm CSSSS with clear shell layer interfaces or eliminate the possibility of elemental diffusion between adjacent layers, which are ascribed to the uniform NaGdF₄ matrix with only slight differences of each shell layer in low amounts of doping ions (e.g., Er ~2%, Nd ~5%, Yb ~5%), we believe that a penta-layer core-shell structure is credible, at least for the excellent ultra-wideband-responsive photon conversion (yet a clear CSSSS should be obtained by using differentiated matrix in future).

- 1) The Er-NCs were synthesized through a typical hot injection method to fabricate core-shell LnNCs. The size of NCs increased gradually from 13.1 to 22.1, 33.9, 40.3, and 41.9 nm (Supplementary Figs. 12 and 16) with the sequential synthesis from C to CS, CSS, CSSS, and CSSSS, accompanied with an excitation spectra evolution of Er-NCs (Fig. 2d), which provided a precondition to forming a core-shell structure.
- 2) The EDS results in Supplementary Fig. 18 showed a gradual appearance of Ho in C, Ho, Gd, Yb in CS, Ho, Gd, Yb, Er in CSS, Ho, Gd, Yb, Er, Nd in CSSS, and Ho, Gd, Yb, Er, Nd in CSSSS with corresponding distributions over the cross-section of Er-NCs, which confirmed a sequential shell growth of NCs from the core. Meanwhile, from the STEM-EDS results in Fig. 2b,c, it could also be found that Ho, Er had a higher abundance near the core section, Gd had a higher abundance in the shell sections, and the low abundance of Yb, Er, Nd exhibited symmetric distribution in Er-NCs.

- 3) The replenished electron energy loss spectroscopy (EELS) analysis further confirmed the distinct abundance and distribution of Ho and Gd, but still failed to outline the exact distribution of Yb, Er, and Nd (Supplementary Fig. 21 and Table II), which was consistent with the result that only doping elements with a ratio higher than 15% could be detected to distinguish core-shell structure in a NaGdF₄ matrix (*Nat. Mater.* **2011**, 10, 968–973)¹.
- 4) Er-based NCs prepared with simply combination of various elements in one matrix or with asymmetric or isolated sensitized structures were found to have worse photo conversion performance than Er-NCs (Supplementary Figs. 30-32), indicating that the designed penta-layer core-shell structure fabrication process was rational.

Overall, based on the above results and previous literature on core-shell LnNCs, we think that the penta-layer core-shell structure (at least a quasi one considering that the diffusion was not excluded) was credible.

Revisions: Supplementary Fig. 21 has been added to the supplementary file, with the corresponding description updated in the manuscript to be:

“...EDS mapping, line scan analyses, and electron energy loss spectroscopy (EELS) analyses consistently revealed the clustered distributions of lanthanide ions (especially the high-amount Gd, Ho, and Er) in different regions of a single nanocrystal (Fig. 2b,c and Supplementary Fig. 21), which provided rational evidence to support the designed penta-layer core-shell structure (Fig. 1c).”

Supplementary Fig. 21 | Electron energy loss spectroscopy spectra collected from spot 1 to 23 on the cross section of a single Er-NC. The index of elements is based on Table I.

Table I | Energy loss of Nd, Gd, Ho, Er and Yb. (Data from <https://eels.info/atlas>)

Major edges	Energy Loss (eV)				
	Nd	Gd	Ho	Er	Yb
M ₄	1000	1217	1392	1453	1576
M ₅	978	1185	1351	1409	1528

Table II | Energy loss intensity of Gd, Ho, Nd, Yb, and Er of a string of spots along the cross section of a single nanocrystal.

	Gd	Ho	Nd	Yb	Er
Spot #1	34.04	4.07	5.33	6.46	7.83
Spot #2	115.30	14.18	1.24	13.93	32.84
Spot #3	111.73	13.28	2.04	6.49	8.38
Spot #4	102.41	10.61	5.36	21.85	13.51
Spot #5	129.19	13.83	0.54	15.41	10.64
Spot #6	97.39	11.61	19.03	7.63	6.24
Spot #7	101.71	9.99	4.88	21.41	12.94
Spot #8	114.81	13.72	0.94	13.59	32.41
Spot #9	74.85	37.91	2.61	15.52	16.32
Spot #10	69.17	28.06	1.31	11.40	12.78
Spot #11	78.47	25.78	-0.05	10.01	12.44
Spot #12	80.15	30.08	-0.12	12.31	8.73
Spot #13	56.44	14.87	-3.78	6.19	4.98
Spot #14	65.86	6.71	3.49	3.29	2.73
Spot #15	76.77	11.02	4.64	6.27	8.78
Spot #16	82.09	8.51	5.98	9.74	5.61
Spot #17	88.66	10.24	1.77	8.01	11.44
Spot #18	88.52	7.46	-0.66	13.45	14.52
Spot #19	61.10	7.98	3.62	7.69	5.96
Spot #20	95.78	15.84	10.62	11.38	11.24
Spot #21	102.25	12.47	1.01	16.02	12.95
Spot #22	129.53	14.14	0.75	15.64	10.93
Spot #23	46.95	5.11	1.75	6.72	6.96

Comment 2. How can it be evidenced for the critical role of the shell-core nanostructure in the photon conversion properties, as compared with that of the simply combination of various elements? How about the material with same elemental doping but without forming the special core-shell nanostructure? Some more discussions should be needed.

Response: We appreciate the reviewer's proposal to compare photon conversion properties between Er-NCs and other Er-based NCs with "simply combination of various elements". In fact, to support the critical role of our designed penta-layer core-shell structure of Er-NCs, we have previously fabricated Ho³⁺, Yb³⁺, Nd³⁺-co-doped NaGdF₄ (Supplementary Fig. 8) and Er-based NCs with asymmetric or isolated sensitized multi-layer core-shell structures (Supplementary Fig.

31 and 32). The former confirmed undesired ET between Ho^{3+} and Nd^{3+} when in a same matrix, while the latter showed obvious weaker photon conversion efficiency.

As suggested, we herein prepared another kind of Er-based NCs with sensitizer, mediator, and activator ions simply combined within one shell. Core-shell $\text{NaGdF}_4@ \text{NaGdF}_4:\text{Ho}^{3+}, \text{Nd}^{3+}, \text{Yb}^{3+}, \text{Er}^{3+}$ NCs were synthesized from a 6-nm NaGdF_4 core (random and ~30-nm sheet-like structure formed in the absence of the core) to guarantee the same morphology and crystal structure as that of Er-NCs. The doping amounts of Gd^{3+} , Ho^{3+} , Nd^{3+} , Yb^{3+} , and Er^{3+} in the NaGdF_4 shell were controlled by using Na/Gd/Ho/Nd/Yb/Er-trifluoroacetates with exact molar ratios of Gd (82.12%), Ho (5.78%), Nd (1.51%), Yb (5.98%), and Er (0.54%) determined from the ICP results of Er-NCs. The core-shell $\text{NaGdF}_4@ \text{NaGdF}_4:\text{Ho}^{3+}, \text{Nd}^{3+}, \text{Yb}^{3+}, \text{Er}^{3+}$ NCs were allowed to grow until they have the same size as that of Er-NCs (41.0 v.s 41.9 nm) before used for photon conversion evaluation.

$\text{NaGdF}_4@ \text{NaGdF}_4:\text{Ho}^{3+}, \text{Nd}^{3+}, \text{Yb}^{3+}, \text{Er}^{3+}$ NCs were found to have notably reduced fluorescence intensity compared to Er-NCs (Supplementary Fig. 30), which also provided a good complement to our conclusion that the strategy and process to prepare CSSSS are feasible (together with the results of Er-based NCs with asymmetric or isolated sensitized multi-layer core-shell structures in Supplementary Fig. 31 and 32).

Revisions: Supplementary Fig. 30 has been added to the supplementary file, with the corresponding description updated in the manuscript to be:

“... which were found to be superior to that of NCs with elements co-doping in one matrix (Supplementary Fig. 30) or with asymmetric or isolated sensitized structures (Supplementary Figs. 31 and 21).”

Supplementary Fig. 30 | **a**, TEM image and the size distribution of co-doping NCs. **b**, Emission spectra of co-doping NCs and CSSSS NCs under the excitation of 448 nm. **c**, Emission spectra of co-doping NCs and CSSSS NCs under the excitation of 793 nm.

Comment 3. Besides the doping ratio, what are the possible effects of the shape, size and impurities on the photon conversion of different CSSSS? And what is the possible effect of the powder size on the elemental distribution (formation of CSSSS)

Response: We thank the reviewer for pointing out these factors that might bring differences to the photo conversion performance between different CSSSS. As suggested, we have analyzed the possible effects of these factors based on current results as follows:

- 1) Doping ratio: the amounts and ratios of doped Ln ions are widely known as the leading factors that determine the excitation and emission behaviors of LnNCs², especially when they functioned differently. In our case where Ho³⁺ and Nd³⁺ were adopted as sensitizers, Yb³⁺ as mediators, and Er³⁺ as activators, the most concern was to achieve ultrawide excitation bands of CSSSS with similar excitation intensity of each characteristic band contributed by Ho³⁺ and Nd³⁺. Our results showed that the doping ratios between Nd³⁺ and Ho³⁺ (Supplementary Fig. 8, 14, and 15), Yb³⁺ and Ho³⁺ (Supplementary Fig. 9), Yb³⁺ and Nd³⁺ (Supplementary Fig. 10), Yb³⁺ and Er³⁺ (Supplementary Fig. 6), and the total ratio of Er³⁺/sensitizers (Supplementary Fig. 17) indeed greatly affected the intensities of excitation and emission of CSSSS. Thus we optimized the amounts of precursors for CS, CSS, CSSS,

and CSSSS (Supplementary Table 1) to give the best doping ratios for desired excitation behaviors and photon conversion performance.

- 2) Shape: the shape or morphology of core-shell LnNCs is generally connected to the crystal structure³. CSSSS were characterized to be high-quality β -NaGdF₄ phase structures (Supplementary Fig. 20). Therefore, while we acknowledge that a quantitative evaluation on this issue would be of a great challenge at the current stage, it is unlikely that a big difference between different CSSSS (as well as CS, CSS, and CSSS) in photon conversion would exist.
- 3) Size: although LnNCs with a larger size generally show stronger fluorescence intensity due to the reduced crystal defects, the vast majority of (multilayer core-shell) NaLnF₄ NCs were reported with tens-of-nanometer sizes to allow safety for bioimaging or dispersibility of the ink for inkjet printing^{4,5}. In our work, CSSSS was also synthesized through the typical protocols and thus was yielded with a similar size of 41~42 nm. Given the high homogeneity (concentrative size distribution) of NCs, we deduce that subtle differences in size would also not produce obvious effects on the photon conversion of different CSSSS.
- 4) Impurities: CSSSS was synthesized in one pot with the gradual injection of each precursor, it is reasonable that the NCs should have the same composition with no more impurities induced between different CSSSS to affect photon conversion performance.

Overall, although CSSSS was carefully fabricated with uniform phase structure and chemical composition by the typical injection method in our work, the factors of doping amounts, shape, size, or impurities would indeed bring some influences to photon conversion by changing the intensities of excitation or emission bands. Luckily, we have managed to obtain the optimal parameters and CSSSS to confirm the ultra-wideband-responsive photon conversion of LnNCs.

As for the possible effect of the powder size on the elemental distribution (formation of CSSSS): to avoid atomic diffusion between each layer, we optimized precursor amounts and optimized reaction parameters (Supplementary Methods) based on typical protocols of core-shell LnNCs. Considering a five-step synthesis was adopted, it was not easy to further reduce the powder size of CSSSS. Meanwhile, it was also found that completely inhomogeneous morphological NCs would generate when a longer retention time at the reaction temperature was allowed in the presence of precursors. Therefore, although we currently have no techniques to testify about the inexistence of even doping distribution or atomic-scale diffusion, we find that the overall performance of CSSSS is satisfying (which might indicate even elemental distribution in CSSSS in reverse).

Comment 4. The thermal stability of the CSSSS materials should be evaluated and discussed for the photon conversion application.

Response: We appreciate this insightful suggestion by the reviewer, as it offers additional opportunities for us to comprehensively understand the performance of LnNCs for wide-ranging applications. The CSSSS materials had good structural thermostability, as they were prepared at ca. 320 °C.

As suggested, we then evaluated the temperature-dependent optical stability of CSSSS by monitoring the photoluminescence spectra of Er-NCs with an 808-nm laser and a 980-nm laser as the excitation sources, respectively. As shown in Supplementary Fig. 24, the emission intensity increased by $\sim 30\%$ with the decrease of temperature from 300 K to 80 K, while decreased by $\sim 48\%$ when the temperature increased from 310 K to 570 K (maintained $\sim 80\%$ at 390 K). Such a phenomenon can be attributed to the increased activity of phonons that leverages the non-radiative relaxation pathways⁶.

Therefore, for applications performed at physiological (e.g., nanotheranostics) or room temperature (e.g., anticounterfeiting) that have no strict demands on fluorescence stability, these CSSSS materials well meet the needs. While for some temperature-monitoring applications (e.g., luminescence thermometry)⁷, the CSSSS materials can provide an additional bonus of temperature-dependent optical properties. However, for some extreme high-temperature conditions, these CSSSS materials might not be a good choice due to the decreased photon conversion efficiency.

Revisions: Supplementary Fig. 24 is added to the supplementary file with the corresponding description updated in the manuscript section of “Ultra-wideband-responsive photon conversion of LnNCs” to be:

“...with temperature-dependent optical stability (Supplementary Fig. 24).”

Relevant measurement methodology has also been provided in the “Optical characterizations” part in Supplementary Methods.

Supplementary Fig. 24 | **a**, Temperature-dependent photoluminescence maps under the excitation of an 808-nm laser. **b**, Intensity of Er emission (1527 nm) in **a**. **c**, Temperature-dependent photoluminescence maps under the excitation of a 980-nm laser. **d**, Intensity of Er

emission (1527 nm) in **c**. (Note that the measurements of the high-temperature part and low-temperature part were performed in different optical paths.)

Reviewer #2:

The manuscript by Zhao et al. reports a type of co-sensitized LnNCs for WLEDs-mediated whole-body angiography of mice and excitation-regulated encryption. The authors have performed many control experiments, which make their experimental findings of robust low-powered WLED pumped LnNCs-based emission very convincing. The result is very interesting and updates the progress in the application of lanthanide materials excited by ultralow-cost light sources. However, I still have some concerns about the novelty or significance of this work. I expect something more than one application.

Response: We are grateful to the reviewer for the positive affirmation of our work “is very interesting and updates the progress in the application of lanthanide materials excited by ultralow-cost light sources”.

In terms of optical applications, besides for use in animal imaging and information encryption, we have also noticed that the feature of directly converting short-wavelength light into second NIR fluorescence of CSSSS can provide huge potentials for chemiluminescence/bioluminescence-based imaging *in vivo*. However, these strategies currently require two or three steps to achieve energy conversions from chemiluminescence/bioluminescence to NIR fluorescence^{8,9}. We currently have demonstrated the construction of a self-illuminating NIR bioluminescent probe by combining LnNCs with a luciferase, NLuc¹⁰. The probe is expected to achieve direct photon conversion from bioluminescence to NIR-II luminescence with only one-step of energy transfer (Fig. I). Meanwhile, we have also found that the multiband response of Er-NCs matched well with the solar spectral irradiances in the range from UV to NIR (Fig. II), indicating that these NCs might also be used for solar conversion¹¹ or solar-powered laser¹².

Besides, we expect that the feature of the distinctive two-process transient spectra of Er-NCs (Fig. 3c,d) might be useful for some applications like fluorescence-lifetime-related imaging technique, however, we currently could not provide more designs.

Fig. I | Illustration for the construction of NIR bioluminescence probe.

Fig. II | Comparison of the excitation spectrum of Er-NCs (em=1527 nm) and the solar spectral irradiances (ASTM G173-03).

There are some specific comments and suggestions:

Comment 1. On page 6, Fig. 3. It maybe not be clear enough for the transient spectrum of NaGdF₄:Ho (ex643 nm). Actually, it exhibits a fast rise-decay profile. However, it seems to be a straight line in Fig. 3c. In addition, the recombination processes from the Nd³⁺→Nd³⁺ and Ho³⁺→Ho³⁺ transitions should be presented in Fig. 3b in terms of solid arrows.

Response: We appreciate the reviewer for these suggestions. We have edited Fig. 3c by bringing the transient spectrum of NaGdF₄:Ho (ex 643 nm) to front layer to avoid the shelter from other curves. Meanwhile, we have also revised Fig. 3b to illustrate the recombination processes from Nd³⁺→Nd³⁺ (⁴F_{3/2}→⁴I_{9/2}; ⁴F_{3/2}→⁴I_{11/2}; ⁴F_{3/2}→⁴I_{13/2}) and Ho³⁺→Ho³⁺ (⁵F₄, ⁵S₂→⁵I₈; ⁵I₅→⁵I₇).

Revisions: Fig. 3 has now been updated to present the profile of NaGdF₄:Ho (ex 643 nm) and to reflect the correct recombination processes.

Fig. 3 | Ultra-wideband-responsive photon conversion of LnNCs. **b**, ET paths of Er-NCs under the excitations of 643 and 574 nm (emission of Yb³⁺ at 980 nm was left out for clarity). **c**, Transient spectra of Er-NCs under the excitation of 643 and 980 nm and of NaGdF₄:Ho under

643 nm excitation. **d**, Transient spectra of Er-NCs under the excitation of 574 and 980 nm and of NaGdF₄:Nd under 574 nm excitation.

Comment 2. On page 22, Supplementary Fig. 27b. According to the description in the main text, the emission spectra of NaGdF₄:Ho which is excited by the monochromic LED at the wavelength of 448 nm should be changed to that being excited by 643-nm LED.

Response: We thank the reviewer for catching this issue. After a careful check, we find that the emission spectrum of NaGdF₄:Ho in Supplementary Fig. 29b (original Supplementary Fig. 27b) is previously mislabeled as 448 nm, and we have now corrected it with the right excitation wavelength of 643 nm (which is a characteristic absorption of Ho³⁺) in the supplementary file.

Revisions: Supplementary Fig. 29 has now been updated to show the coherent excitation condition of NaGdF₄:Ho (643 nm).

Supplementary Fig. 29 | a, Excitation spectra of the as-prepared Er-NCs, NaGdF₄:Ho³⁺, and NaGdF₄:Nd³⁺. **b**, Emission spectra of the as-prepared Er-NCs, NaGdF₄:Ho³⁺, and NaGdF₄:Nd³⁺.

Comment 3. Regarding the whole-body angiography experiment, NaHoF₄@NaGdF₄:Ho/Yb@NaGdF₄:Yb/Er@NaGdF₄:Nd/Yb@NaGdF₄ was utilized as the sample agent for its intense NIR-II emission at 1527 nm (Er³⁺, ⁴I_{13/2}-⁴I_{15/2}). In the main text, the authors have performed real-time whole-body angiography of mice under the excitation of 808-nm laser and a WLED, respectively. It is verified that the performance of Er-NCs excited by WLED is superior to that of the 808-nm laser pumped case (SFig.37). It is not reasonable for this experimental settings. Since the 808-nm energy can only be absorb by Nd³⁺ ions, and then transferred to Er³⁺ by Yb-sublattice, while there are no response from the NaHoF₄@NaGdF₄:Ho/Yb@NaGdF₄:Yb/Er part at all under the excitation of 808-nm laser. Hence, the author should conduct the comparative tests for Er-NCs under the excitation of a WLED and 980-nm laser (instead of 808-nm one), particularly the

adsorptivity data of WLED and 980-nm laser. Note that, the heating effect can be ignored when the pumping density of 980-nm laser diode is clamped at $\sim 0.12 \text{ W cm}^{-2}$.

Response: We thank the reviewer for this constructive suggestion. We encountered the same situation when performing whole-body angiography with Er-NCs.

The initial choice of using the 808-nm laser, other than the 980-nm one, for comparison with a WLED was made because we focused mostly on the contributions of the two sensitizers of Ho^{3+} and Nd^{3+} to excitation modulation, considering that Ho^{3+} and Nd^{3+} had multiple characteristic absorption peaks within 300-900 nm, while neither of them showed obvious absorption around 980 nm (Supplementary Fig. 2). The exact absorption behavior was further confirmed by the quantitative absorption cross-section calculation of Ho^{3+} and Nd^{3+} within the range of 300 to 1100 nm (Supplementary Fig. 41). We thus deduced that the superior performance (absorptivity) to that of the widely used 808-nm laser by a WLED should endow Er-NCs with comparable or at least worth-trying potentials when used for bioimaging, and the whole-body angiography outcomes indeed showed that Er-NCs were excitable by both light sources with high SNR to outline clear vascular structure (Fig. 4f,g).

The 980-nm laser was not adopted because it could barely be absorbed by either sensitizer of Ho^{3+} or Nd^{3+} , but could be absorbed by the mediator of Yb^{3+} (Supplementary Fig. 2 and Supplementary Fig. 41), which was beyond the scope of our co-sensitization strategy and multi-layer core-shell structural design. Besides, $\text{Yb}^{3+}, \text{Er}^{3+}$ -co-doped LnNCs have also been widely demonstrated with excellent performance in bioimaging and cancer treatment^{13,14}.

Here as suggested, we have also conducted additional whole-body angiography experiments. As expected, more refined vascular structures were visualized with higher SNR upon excitation by a 980-nm laser (Supplementary Fig. 40 and Supplementary Video 4), compared to that of a WLED or an 808-nm laser.

Overall, the scope of this manuscript is to introduce an efficient co-sensitization strategy to achieve ultra-wideband absorption of LnNCs. And we aim to demonstrate that the excitation sources for high-resolution bioimaging can be extended to easily available white light beyond typical NIR lasers while maintaining acceptable imaging resolution (other than to screen out the best imaging sources out of the WLED, 808-nm laser and 980-nm laser).

Revisions: Supplementary Fig. 2 has been updated to extend the wavelength from 300 to 1100 nm. Supplementary Fig. 40 has been added to the supplementary file with the corresponding description updated in the manuscript to be:

"...A higher SNR of 19.0 was also obtained with a 980-nm laser (Supplementary Fig. 40 and Supplementary Video 4), which might be attributed to the strong absorbance at 980 nm by the mediator of Yb in Er-NCs. Due to the high absorptivity of Er-NCs to WLED (0.3515 m^2), 808-nm laser (0.2687 m^2) and 980-nm laser (0.9867 m^2) (Supplementary Fig. 41), the excitation sources of LnNCs for high-resolution bioimaging could therefore be extended to visible light beyond previously reported sole NIR lasers or low-powered LED."

Supplementary Fig. 41 has also been updated with discussions on the absorptivity of Er-NCs to the WLED, 808-nm laser, and 980-nm laser.

Supplementary Fig. 2 | Absorption spectra of ErCl_3 , HoCl_3 , NdCl_3 , PrCl_3 , TmCl_3 and YbCl_3 in aqueous solution (0.2 M).

Supplementary Fig. 40 | Whole-body angiography of a mouse using a 980-nm laser as the excitation source, and the analysis on cross-sectional intensity (black line) and Gaussian fit fluorescence intensity profiles (cyan line).

Supplementary Fig. 41 | a-d, Absorption cross-section of a, Ho^{3+} , b, Er^{3+} , c, Nd^{3+} , and d, Yb^{3+} . e, Calculated absorption cross-section of 1 mol of Er-NCs. f, Radiation spectra of the adopted 808-nm laser, 980-nm laser, and WLED. g, Principle for the calculation of absorptivity of Er-NCs to the specific light source.

Comment 4. The absorption spectra of Ln^{3+} -NCs ($\text{Ln} = \text{Er}, \text{Tm}, \text{Pr}, \text{Ho}$) are quite important to assess their performances. However, i cannot find those data until the calculated absorption cross-section of Er-NCs in Supplementary Fig. 37.

Response: We appreciate the reviewer for this comment. As suggested, we have measured the absorption spectra of Er/Ho/Pr/Tm-NCs and deposit these spectra in Supplementary Fig. 26 to support a mutual confirmation with their respective excitation spectra.

Revisions: Supplementary Fig. 26 has been updated with new additional absorption spectra of Er/Ho/Pr/Tm-NCs with the corresponding description in the revised supplementary file.

Supplementary Fig. 26 | a, Absorption spectra of Er/Ho/Pr/Tm-NCs.

Reviewer #3:

This manuscript describes a heterogenous design of rare earth doped nanoparticles that can absorb the broad band LED light and emit both upconversion in visible and down-shifting luminescence in near infrared second window for animal imaging and anticounterfeiting (data storage) applications. The investigation is comprehensive with a vast amount of synthesis and systematic characterizations of the materials as well as two directions of application demonstrations. Most of the data here are convincing and coherent. The strategy of co-sensitization by employing Nd^{3+} and Ho^{3+} as the co sensitizers with complementary absorptions from the ultraviolet to infrared are convincing. The formula of $20\%\text{Ho}^{3+}/1\%\text{Yb}^{3+}$ and $5\%\text{Nd}^{3+}/5\%\text{Yb}^{3+}$, designed in the penta-layer core-shell (core/shell/shell/shell/shell, CSSSS) structure can provide an excellent reference for the community to repeat the experiments. Remarkably, the low-powered white LED illumination suggests the potentials in the whole-body angiography of mice. I do not have other major comments except that the illustration of Figure 1 b was not consistent to the actual design of the nanoparticles, which I suggest this part of Figure should be re-drawn.

Response: We are grateful to the reviewer for the positive affirmation of our work. As suggested, we have redrawn Fig. 1b with a universal ultra-wideband-responsive nanostructure that contains multiple optical units of the *co-sensitizer-mediator-activator triplet*, based on the co-sensitization mechanism (Fig. 1a) and the model symmetric penta-layer core-shell CSSSS (Fig. 1c), which we wish to be satisfying. Besides, Supplementary Fig. 1 has also been updated to be consistent.

Revisions: Re-drawn Fig. 1b has been updated in Fig. 1 in the revised manuscript to reflect good consistency with the actual design. Supplementary Fig. 1 has also been updated in the revised supplementary file.

Fig. 1 | Schematic illustration of co-sensitization strategy and design of symmetric multilayer core-shell structure. a, Mechanism of the ET within a co-sensitization structure. **b,** Illustration of a universal ultra-wideband-responsive structure with multiple optical units. **c,** Illustrated cross-section of prepared CSSSS with symmetric penta-layer core-shell structure (X=Er, Ho, Pr, and Tm).

Supplementary Fig. 1 | Schematic illustration and tuning of sensitization strategies. a-c, Illustration of the ET within a sensitizer-activator pair (**a**), a single sensitization structure (**b**), and the proposed co-sensitization structure (**c**).

References

1. Wang, F. et al. Tuning upconversion through energy migration in core–shell nanoparticles. *Nat. Mater.* **10**, 968–973 (2011).
2. Zheng, B. et al. Rare-earth doping in nanostructured inorganic materials. *Chem. Rev.* **122**, 5519–5603 (2022).
3. Mai, H.-X. et al. High-quality sodium rare-earth fluoride nanocrystals: controlled synthesis and optical properties. *J. Am. Chem. Soc.* **128**, 6426–6436 (2006).
4. Zhou, J., Liu, Q., Feng, W., Sun, Y. & Li, F. Upconversion luminescent materials: advances and applications. *Chem. Rev.* **115**, 395–465 (2015).
5. Zhou, B. et al. NIR II-responsive photon upconversion through energy migration in an ytterbium sublattice. *Nat. Photonics* **14**, 760–766 (2020).
6. Zhou, J. et al. Activation of the surface dark-layer to enhance upconversion in a thermal field. *Nat. Photonics* **12**, 154–158 (2018).
7. Brites, C. D. S., Balabhadra, S. & Carlos, L. D. Lanthanide-based thermometers: at the cutting-edge of luminescence thermometry. *Adv. Opt. Mater.* **7**, 1801239 (2019).
8. Lu, L. et al. NIR-II bioluminescence for in vivo high contrast imaging and in situ ATP-mediated metastases tracing. *Nat. Commun.* **11**, 4192 (2020).
9. Yang, Y. et al. NIR-II chemiluminescence molecular sensor for in-vivo high contrast inflammation imaging. *Angew. Chem. Int. Ed.* **59**, 18380–18385 (2020).
10. Hall, M. P. et al. Engineered luciferase reporter from a deep sea shrimp utilizing a novel imidazopyrazinone substrate. *ACS Chem. Biol.* **7**, 1848–1857 (2012).
11. Hadke, S. et al. Emerging chalcogenide thin films for solar energy harvesting devices. *Chem. Rev.* **122**, 10170–10265 (2022).
12. Graham-Rowe, D. Solar-powered lasers. *Nat. Photonics* **4**, 64–65 (2010).
13. Zhong, Y. et al. Boosting the down-shifting luminescence of rare-earth nanocrystals for biological imaging beyond 1500 nm. *Nat. Commun.* **8**, 737 (2017).
14. Zhong, Y. et al. In vivo molecular imaging for immunotherapy using ultra-bright near-infrared-IIb rare-earth nanoparticles. *Nat. Biotechnol.* **37**, 1322–1331 (2019).

REVIEWERS' COMMENTS

Reviewer #1 (Remarks to the Author):

The authors' replies are reasonable to me and revisions are useful. I think the paper is proper for publication.

Reviewer #2 (Remarks to the Author):

The authors have clearly answered all my questions. Now I can confirm the novelty and the potential impact of this research. I believe this manuscript is ready for publishing on Nature Communications.

Author Rebuttals to First Revision:

Reviewer #1:

The authors' replies are reasonable to me and revisions are useful. I think the paper is proper for publication.

Response: We are thankful to the reviewer for their thoughtful comments and time in reviewing our manuscript, which have resulted in marked improvements to the manuscript.

Reviewer #2:

The authors have clearly answered all my questions. Now I can confirm the novelty and the potential impact of this research. I believe this manuscript is ready for publishing on Nature Communications.

Response: We are grateful to the reviewer for their valuable time and expertise in reviewing our manuscript, which have resulted in marked improvements to the manuscript.